# "Where am I?" A snapshot of the developmental topographical disorientation among young Italian adults

**Laura Piccardi**[1,2]*, **Massimiliano Palmiero**[3], **Vincenza Cofini**[4], **Paola Verde**[5]*, **Maddalena Boccia**[1,6], **Liana Palermo**[7], **Cecilia Guariglia**[1,6], **Raffaella Nori**[8]

**1** Department of Psychology, "Sapienza" University of Rome, Rome, RM, Italy, **2** IRCCS San Raffaele, Rome, RM, Italy, **3** Department of Biotechnological and Applied Clinical Sciences, University of L'Aquila, L'Aquila, Italy, **4** Department of Life, Health and Environmental Science, University of L'Aquila, L'Aquila, Italy, **5** ItAF Aerospace Test Division, Aerospace Medicine Department, Pratica di Mare, RM, Italy, **6** Cognitive and Motor Rehabilitation and Neuroimaging Unit, IRCCS Fondazione Santa Lucia, Rome, RM, Italy, **7** Department of Medical and Surgical Sciences, Magna Graecia University of Catanzaro, Catanzaro, Italy, **8** Department of Psychology, University of Bologna, Bologna, BO, Italy

* laura.piccardi@uniroma1.it (LP); paola.verde@aeronautica.difesa.it (PV)

**Data Availability Statement:** All data collected are available at the following link: https://osf.io/qwapm/?view_only=

## Abstract

In the last decade, several cases affected by Developmental Topographical Disorientation (DTD) have been described. DTD consists of a neurodevelopmental disorder affecting the ability to orient in the environment despite well-preserved cognitive functions, and in the absence of a brain lesion or other neurological or psychiatric conditions. Described cases showed different impairments in navigational skills ranging from topographic memory deficits to landmark agnosia. All cases lacked a mental representation of the environment that would allow them to use high-order spatial orientation strategies. In addition to the single case studies, a group study performed in Canada showed that the disorder is more widespread than imagined. The present work intends to investigate the occurrence of the disorder in 1,698 young Italian participants. The sample is deliberately composed of individuals aged between 18 and 35 years to exclude people who could manifest the loss of the ability to navigate as a result of an onset of cognitive decline. The sample was collected between 2016 and 2019 using the Qualtrics platform, by which the Familiarity and Spatial Cognitive Style Scale and anamnestic interview were administered. The data showed that the disorder is present in 3% of the sample and that the sense of direction is closely related to town knowledge, navigational strategies adopted, and gender. In general, males use more complex navigational strategies than females, although DTD is more prevalent in males than in females, in line with the already described cases. Finally, the paper discusses which protective factors can reduce DTD onset and which intervention measures should be implemented to prevent the spread of navigational disorders, which severely impact individuals' autonomy and social relationships.

2da5876783c842779a4d1a9380784688 as reported also in the paper.

**Funding:** The authors received no specific funding for this work.

**Competing interests:** The authors have declared that no competing interests exist.

## Introduction

Navigation is the ability to move from one location to the next, following habitual routes and avoiding getting lost [1] in new and familiar environments. Due to its importance, it is not surprising that many cognitive processes (i.e., memory, mental imagery, attention to landmarks and other features, and our perception of directions and distances as well as decision-making, planning, and problem-solving) [2–6], and multiple brain regions [7–11] are involved in successful navigation. Specifically, the success of this process also depends on internal and external factors to the individual. The internal factors are of greater interest because they directly affect navigational competence; by consequence, it is possible to better intervene on them in order to plan prevention programs related to navigational disorders. The most important internal factors for navigation are: a) the cognitive predisposition to grasp certain environmental information rather than others [12–16]; b) gender [17–24]; c) age [25–30]; d) professional experience [31–36]; e) familiarity with the environment [37–40], reflecting the result of repeated exposure to a stimulus or an environment [41, 42]; f) navigational strategies used during navigation [36, 43]; g) psychiatric (e.g., spatial anxiety, depression, agoraphobia: [44–46]) and neurologic diseases (Alzheimer's disease and brain lesions in the navigational brain network: [47–50]).

In the present study we focused on an indirect measure of navigation, namely the Sense of Direction (SOD), which is our own perception about navigation ability, in order to understand the critical internal variables that affect it: the demographic factors (e.g., age, gender and education), the degree of familiarity with the environment (e.g., town knowledge), and navigational strategies, which include not only cognitive styles (e.g., landmark, route and survey), but also the preferential mean to explore the environment, that is to say, means of travelling (MoT: active, passive). In addition, we also considered the right-left confusion (RLC), with reflects a pathological condition that can be associated or not with navigational disorders.

At the aim, Siegel and White's model [51] is used as the theoretical framework. This model suggests that environmental knowledge occurs in three distinct hierarchical steps, namely Landmark (i.e., figurative memory of environmental objects), Route (i.e., a sequence memory of the path connecting environmental objects in an egocentric perspective), and Survey (i.e., map-like representation in an allocentric perspective) knowledge. Following this model, people can be classified in three categories according to the navigational strategies they adopt: landmark (less skilled and get lost easily), route (more skilled at connecting landmarks by verbal labels, such as right, left, behind, ahead, etc.., but less able at finding shortcuts or changing routes after environmental changes), and survey (the most navigational skilled, excellent visual-spatial abilities) users [52, 53]. Notably, both egocentric (route representations) and allocentric (survey representations) frames of reference are needed to specify categorical (nonmetric) and coordinate (metric) spatial encodings, which are two different but complementary aspects of spatial cognition [54–56]. According to Montello [5], even though familiarity (the repetitive exposure to a particular environment) is more associated with a survey format, spatial knowledge would be not hierarchically organized, but rather would occur in a parallel fashion, depending on the situation [57]. This means that navigation is very complex and that different difficulties can arise when moving through the environment. Furthermore, in the Environmental Knowledge Model [37] emerges as the environmental familiarity allows to perform navigational tasks requiring higher spatial cognitive strategies (e.g. environmental map; perspective changing) even in those individuals with poor spatial orientation skills. In this vein, familiarity with environment enhances wayfinding and reduces wandering and topographical disorientation in individuals suffering from neurocognitive decline [39, 58]. The

effect due to familiarity emerges also in healthy ageing given that older people may be better than younger people [59].

Amongst others, the right-left confusion can explain same individual differences in term of SOD [60–62]. Indeed, the ability to discriminate left from right from one's own perspective is useful when individuals explore and recall the environment [60]. In addition, the ways of travelling can also affect the ability to orient oneself in space. Specifically, active travel (e.g., moving by car or bike or on foot) often indicates a better SOD and leads to a better representation of one's surroundings [63], whereas passive travel (e.g., moving as a passenger in a car, but, taxi, etc...) is likely related to less navigation ability [64]. Yet, Bocchi et al. [12] found that navigational strategies can also affect SOD. Indeed, landmark users can show more difficulties, whereas survey users are the most skilled in solving navigational problems and in travel planning [53, 65–67].

Most importantly for the purpose of the present study, navigational difficulties can be associated to a neurodevelopmental disorder that specifically undermines navigational skills. This disorder was described for the first time by Iaria et al. [68] and was named 'Developmental Topographical Disorientation (DTD)'. Afterwards, numerous people worldwide have been identified as suffering from this condition [69–75]. Iaria and Barton [74] demonstrated that DTD is widespread in the Canadian population, finding 120 individuals fulfilling the criteria for a diagnosis of DTD. This led to conclude that DTD is rather widespread in the population and requires targeted interventions by clinical services. In general, people with DTD have normal memory and neuropsychological profiles but show a major cognitive deficit in spatial cognition, and complain of severe problems in navigating on an everyday basis. Specifically, they are unable to use cognitive maps or place-based navigation strategies to find their way around familiar and novel environments. In general, individuals with DTD show a higher impairment of metric (coordinates) than nonmetric (categorical) spatial encoding, and, basing on Siegel and White's model, they hardly reach a level of route knowledge of the environment, often stopping at a landmark knowledge, highlighting a lack of allocentric representation capacity [76]. By analysing single cases reports, subjects are characterised by a different degree in terms of severity of illness and in terms of navigational impairment. Specifically, Case one had a severe deficit in the formation of the environment mental map [66]; FG had a normal acquisition of environmental information but a specific impairment in the retrieval with a loss of information after 5 minutes [69]; Dr. WAI and LA had both normal acquisition and retrieval of environmental information [70, 73]. FG, LA, and Dr. WAI had deficits in mental representation, mental rotation and mental generation of environmental images. Nobody had difficulty in landmark recognition. LG, instead, was the first case of Landmark Agnosia Development, showing a selective deficit in recognizing landmarks allowing spatial orientation [77]. CF [75] was fully effective in learning and following routes and in building up cognitive maps as well as in recognizing landmarks. However, she performed significantly worse than age and gender-matched controls on the map-following task, namely when she was required to use a map to navigate in a novel environment. In terms of neural correlates, the first studies on DTD cases showed no clear brain structural abnormalities (i.e., [68, 69]). However, using an fMRI landmark sequencing task, Nemmi and co-workers [71] demonstrated that DTD individuals did not show any activation in the navigation brain network, whereas prefrontal areas, known to be involved in processing the sequential order of everyday life actions [78, 79], were normally activated. The decreased functional connectivity between the hippocampus and the prefrontal cortex has also been described [80] and interpreted as a defective functioning of two crucial areas for navigation and decision making. In addition, the rs-fMRI experiment demonstrated aberrant functional connectivity between regions within the default-mode network

(DMN), specifically between the medial prefrontal cortex and the posterior cingulate cortex, the medial parietal and temporal cortices.

Thus, the objective of the present study was: to estimate the percentage of the DTD among a convenience sample of Italian adults aged 18–35 years to define soon clinical lines of intervention and a protocol of investigation shared on the national network. To this purpose we investigated the SOD and its correlates. We hypothesize that some internal factors may be correlated to the presence of DTD. Specifically, we expect the gender distribution to be different between males and females: in line with previous described cases we expect to find more males suffering from DTD than females [69, 70, 73, 75, 77]. Moreover, we hypothesize that DTD correlates differently to navigational strategies: we expect that individuals with Survey abilities show higher navigational skills [36] and less probability to have DTD symptoms with respect to people with lower navigational skills (Landmark and Route users). In this vein, we also hypothesize that individuals with DTD show higher right/left confusion and a passive use of means of transport. Consistent with the familiarity effect on SOD [37, 39, 40, 58, 59], we also hypothesize that familiarity, measured as town knowledge, may affect SOD favouring navigation, and could be a protective factor for individuals with DTD who may be able to perform certain spatial orientation tasks in a familiar environment.

## Methods

### Study design

For this study we conducted an online survey among Italian young people in order to investigate the presence of the DTD and its individual correlates.

### Study population

**Participants.** The eligible study population consisted of all those people without neurological disorders and with an age between 18 and 35 years. A sample of 1,698 participants took part in the experiment. Participants had a full-time education, ranging from 8 to 18 years (mean = 14.80 years, SD = 2.83 years). They were not all university students. Specifically, 81 (4.79%) participants achieved: a secondary school diploma; 842 (49.76%) a high school diploma; 769 (45.45%) a degree or post-degrees. Demographic data of all participants are reported in Table 1. The study was performed according to the ethical principles expressed in the Declaration of Helsinki and it was approved by the Local Ethics Committee (Department of Psychology, University of Bologna, Italy).

**Table 1. Demographic data of participants.**

| | |
|---|---|
| N. total | 1,698 |
| N. males | 635 |
| N. females | 1,063 |
| Age Total (years) | 24.89 (4.08) |
| Age males (years) | 25.56 (4.25) |
| Age females (years) | 24.48 (3.92) |
| Education Total (years) | 14.80 (2.83) |
| Males Education (years) | 14.39 (2.08) |
| Females Education (years) | 15.04 (2.82) |

Note: means (Standard Deviations).

## Data collection

Participants from all Italian regions (from North to South, including the Islands) were recruited between 2016 and 2019 using notices on social networks and on bulletin boards of researchers. The advices about the survey were basically spread out by word of mouth and flyers, that were distributed in community meeting points, such as bookshops, cafeterias, public library, and sport clubs. The software Qualtrics (First release: 2005, Provo, Utah, USA, Available at: https://www.qualtrics.com) was used. All participants gave their informed consent before their inclusion in the study.

## Measures

**A. Anamnesis questionnaire.**   Participants had to fill in some questions about problems of spatial orientation from an early age, neurological or major psychiatric illness, previous traumatic brain injury, history of learning disabilities, alcohol or drug abuse.

Specifically, with regard to neurological outcomes, participants were asked to specify whether they had experienced head trauma, ischaemic attacks, encephalitis, brain infections, pre-perinatal complications. For the psychiatric side, whether they had suffered or were suffering from depression, anxiety, psychosis, obsessive-compulsive disorder, eating disorder, post-traumatic stress disorder, schizophrenia, phobias. When he/she suffered from it and whether he/she was treated pharmacologically and/or is currently on medication. For substance use, we investigated whether he or she uses alcohol, if so how often, and whether he or she uses or has used drugs (cannabis, amphetamines, cocaine etc.) if so when did he or she use them, how often and which substances.

**B. Familiarity and spatial cognitive style scale [81, 82].**   Participants had to fill in a series of questions: each item of the scale was a self-referential statement about some aspect of environmental spatial cognition. At the beginning of the scale participants were asked to report demographic information (age, gender and educational level), as well as how they moved around the environment, that is whether they used active or passive means of transport. Specifically, we investigated the use of means of transport, distinguishing active means of transport in which the participant actively drives and moves around the environment (i.e., driving a car; riding a moped; riding a bicycle; riding a motorbike; walking), and passive means of transport in which the participant is led around the environment by others (i.e., being a passenger in a car; using a taxi; using a bus; using a train). For each choice of means of transport, the participant indicated on a scale of 1 to 5 how often they used it. Based on the frequency of use of the various means, the prevalence of active or passive use of the means of transport was defined.

In addition, they were also asked to indicate in the section 'town knowledge' to think of a town they knew well even if it was different from the one they lived in.

The scale was divided into the following subscales:

1. Sense of Direction (SOD) was the summed rating for items concerning the sense of direction, Items 1, 2, 3, 4, 6, 7, 9, 10, 11, and 22 (e.g., Item 1: "How is your ability to read a map?");

2. Town Knowledge (TK) was the summed ratings of Items 8, 12, 13, 14, 15, 16, 19, 20, and 21 [e.g., Item 12: "How well do you know (insert the name of the city where you live in)?"];

3. Spatial cognitive style. To evaluate individual spatial cognitive style, two items were used [Items 17 (a, b, c) and 18 (a, b, c)]. (e.g., given the following item "Try to imagine a route you usually take (e.g., home to work, college to cafeteria. . ."), participants were asked to evaluate each of the following strategies: a) Landmark—Do you visualize only the

landmarks (e.g., your home, the cafeteria. . .)? b) Route—Do you visualize both the land-marks and the route leading to your destination? c) Survey—Do you ever imagine the route as if it were on a map?).

4. Right-left confusion (RLC). Confusion of right and left self-referents was obtained by Item 5, "In everyday life, do you confound right and left?".

For each item, participants should circle a number from 1 to 5 to indicate their response: higher numbers correspond to higher ability. In previous works [82–84], the overall value of the Cronbach's alpha for the total scale ranged between .79. and .74, which is good [84], as the test-retest reliability as reported by Nori and Piccardi [81]. In the present sample, Cronbach's alpha for the total scale was = .71. In Piccardi, Risetti, Nori [82] the internal consistency of SOD and TK was estimated as a combined score (.74). In the present study, the internal consistencies of SOD (.73) and TK (.71) were estimated separately. The full *Familiarity and Spatial Cognitive Style Scale* is available in the appendix of Piccardi, Risetti and Nori [82].

In this study, the final scores were the means of the item scores for SOD, TK and spatial cognitive styles (landmark, route and survey), whereas 'right-left confusion' and the 'ways of travelling' were analysed as dichotomous variables. First of all, regarding the item 'right-left confusion' (hereafter RLC) we dichotomised as follows: 'yes' if the subject responded 'some-times', 'often' or 'always'; 'no' if the subject responded 'never' or 'rarely'. Secondly, the ways of travelling, that is the Means of Transport (hereafter MoT) was computed from 5 items: 1) drove a car; 2) other MoT (e.g., bike); 3) only travelled as a passenger; 4) took public transport; 5) only moved on foot. Thus, a subject was defined as an 'active traveller' if he/she indicated to move mainly by car, motorbike, bicycle or on foot; a subject was defined as a 'passive traveller' if he/she indicated to move mainly as a passenger in a car, taxi, coach, bus, train, air, etc.

Specifically, we classified participants with DTD if they reported a total SOD of 2 Standard Deviations (SD) below the mean (95% CI), as calculated from data collected by Nori and Piccardi [79]. Furthermore, on the basis of the anamnesis questionnaire we also considered the four following diagnostic criteria suggested by Iaria and Barton in [74]: i) getting lost daily or often (1 to 5 times a week) in the most familiar environments; ii) the problem of spatial orientation must be present from an early age; iii) no other cognitive difficulties that may affect daily life activities, and as the last criterion; iv) no known brain lesions, malformation or any condition affecting the central nervous system, with the exception of migraine. In addition, we also took into account for two adding criteria: v) no psychiatric disorders and psychotropic drug use, and vi) substance abuse behaviour.

## Statistical methods

Continuous variables were reported as means (Standard Deviation = SD), and categorical factors were reported as percentages. To identify the variables that were significantly related to SOD, the Spearman correlation coefficient was computed for variables measured at least on ordinal scale: Age, Educational Level (low: people with at most a high school diploma; high: degree or post-degrees), TK (score), Landmark (score), Route (score) and Survey (score); and the Point-Biserial Correlation Coefficient for the nominal variables: Gender, RLC (yes/no) and MoT (active/passive). When the correlation coefficients were statistically significant (p < .05) the variables were introduced into a Generalised Linear Regression model (Glm). For the Glm model, an identity link function and a normal family distribution were specified as a linear model, with SOD as the continuous dependent variable and the following model terms as independent variables: Age, TK (score), Landmark (score), Route (score) and Survey (score) as centred covariates; Gender (M/F), RLC (yes/no) and MoT (active/passive) as dummy factors.

All interactions with categorical variables were introduced because they explained more variability of SOD ($R2 = .337$).

In addition, to study the associations between DTD and each socio-demographic variable and each different subscales of the Family *and Spatial Cognitive Style Scale*, the Chi-square test and One-way analysis of variance, for categorical or continuous data respectively, were used. Specifically, the univariate Logistic regression model, taking DTD as binary dependent variable (yes/no) and one independent variable at a time (Gender, Age, Educational level, RLC, MoT, TK, Landmark, Route, Survey) was carried out to estimate significant predictors, reporting the unadjusted odds ratio (OR) with their 95% Confidence Intervals (95% CI). The significant predictors ($p < .05$), estimated by univariate analysis, were introduced into a multivariable logistic model, in order to both estimate the odds ratio controlling for the other covariates (adjusted Odds ratios: ORadj), and identify significant protective or risk factors for DTD.

The analyses were performed by using STATA/MP14 software and the jamovi project (2021): *jamovi* (Version 1.6) Computer-Software, setting alpha to .05. All tests were two-tailed.

## Results

### General characteristics

Participants reported higher means for the Route scale (3.87, SD = 4), the Landmark (3.58, SD = 3.5) scale, and the SOD scale (3.14, SD = .6), while for the TK scale and the Survey scale the mean values were 2.79, SD = 2.9 and 2.55, SD = 2.5 respectively. Fig 1 shows the different score levels measured by the different subscales.

### Correlation between factors investigated and SOD

As reported in Table 2, Spearman correlation analysis showed that SOD was negatively correlated with a weak magnitude level to gender, RLC and Educational level. In addition, SOD was positively correlated with a magnitude ranging from weak to moderate level to Landmark, Route and Survey scores. Moreover, SOD was also positively correlated with a weak magnitude level to Age, MoT and TK. (see [85, 86] for correlation magnitude).

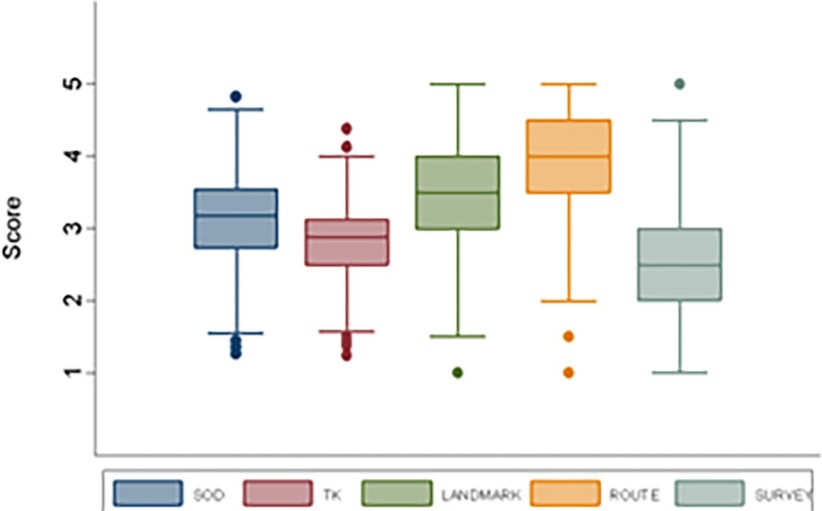

**Fig 1. Box plot of SOD scale and TK, Landmark, Route and Survey subscales.**

**Table 2. Spearman correlation between SOD and investigated factors.**

| Parameter | n | Correlation coefficients (rho) | degrees of freedom | p-value |
|---|---|---|---|---|
| Age (years) | 1698 | .069 | 1696 | .004 |
| Gender (M/F) | 1698 | -.289 | 1696 | < .001 |
| Educational Level (Low/High) | 1692 | -.045 | 1690 | .066 |
| RLC (yes/no) | 1697 | -.163 | 1695 | < .001 |
| MoT (active/passive) | 1392 | .064 | 1390 | .017 |
| TK (score) | 1697 | .290 | 1695 | < .001 |
| Landmark (score) | 1697 | .119 | 1695 | < .001 |
| Route (score) | 1697 | .276 | 1695 | < .001 |
| Survey (score) | 1698 | .506 | 1698 | < .001 |

Note: RLC = Right-Left confusion; MoT = Means of Travelling; TK = Town Knowledge.

Using the generalised linear model, the multivariable analysis showed that SOD was related to gender and spatial abilities. In particular, SOD increased because of Survey ability (Beta coefficient = .241; p < .01), whereas decreased in subjects with RCL (Beta coefficient = -.555; p = .009). The model showed that the investigated interactions were not significant (Table 3).

Fifty-four participants met the criteria for DTD, then the percentage of DTD was 3% of our sample (54/1698: 95% CI: 2.4% - 4.0%). The logistic univariate analysis showed that DTD was associated with gender: the males showed a higher risk than females to have DTD (OR: 2.39; 95% CI: 1.2–4.7; p = .009). The risk of DTD was lower in people with higher scores in TK, Route, and Survey scales (Table 4).

The multivariable logistic regression analysis confirmed that the risk of DTD was statistically lower in subjects with a higher Survey Ability (ORadj = .26; 95% CI: .16 - .40), adjusted for all factors investigated (Fig 2).

## Discussion

The present online investigation aimed to estimate the presence of DTD in a large young Italian sample to better understand the spread of this condition in the population. Thus, we

**Table 3. Factors related with SOD (multivariable analysis).**

| Independent variables | Estimate | SE | Beta | 95%CI lower | 95%CI upper | z | p |
|---|---|---|---|---|---|---|---|
| (Intercept) | 3.225 | .0636 | .098 | -.11 | .30 | 50.670 | < .001 |
| Gender (male) | -.179 | .0827 | -.293 | -.56 | -.03 | -2.157 | **.031** |
| TK | .199 | .0310 | .147 | .10 | .19 | 6.424 | **< .001** |
| Landmark | .055 | .0164 | .078 | .03 | .12 | 3.366 | **< .001** |
| Route | .082 | .0195 | .104 | .06 | .15 | 4.227 | **< .001** |
| Survey | .241 | .0152 | .389 | .34 | .44 | 15.823 | **< .001** |
| MoT (active) | .082 | .0683 | .134 | -.08 | .35 | 1.200 | .230 |
| Age | .001 | .0034 | .002 | -.00 | .01 | .358 | .720 |
| RLC (no) | -.555 | .2122 | -.909 | -1.59 | -.23 | -2.610 | **.009** |
| Gender * MoT | -.045 | .0888 | -.074 | -.36 | .21 | -.510 | .610 |
| Gender * RLC | .425 | .2343 | .697 | -.06 | 1.45 | 1.815 | .070 |
| MoT* RLC | .418 | .2312 | .684 | -.06 | 1.43 | 1.806 | .071 |
| Gender*MoT* RLC | -.777 | .2547 | -.617 | -1.43 | .20 | -1.479 | .139 |

Note: RLC = Right-Left confusion; MoT = Means of Travelling; TK = Town Knowledge.

**Table 4. Factors related to DTD (univariate logistic regression).**

| | DTD | | | | |
|---|---|---|---|---|---|
| | **No (n = 1644)** | **Yes (n = 54)** | **P*** | **OR** | **95%CI** |
| Factors | N (%) or mean (SD) | N (%) or mean (SD) | | | |
| Gender | | | | | |
| Females | 624 (98%) | 11 (2%) | | 1 | |
| Males | 1020 (96%) | 43 (4%) | .009 | 2.39 | 1.2–4.7 |
| Age (yrs.) | 24.9 (4.1) | 24.8 (3.9) | .9429 | 1.00 | .9–1.1 |
| Educational Level | | | | | |
| Low | 895 (97%) | 28 (3%) | | 1 | |
| High | 743 (97%) | 26 (3%) | .686 | 1.1 | .6–1.9 |
| RLC | | | | | |
| No | 1276 (97%) | 37 (3%) | | 1 | |
| yes | 367 (96%) | 17 (4%) | .114 | 1.6 | .9–2.9 |
| MoT | | | | | |
| Passive | 184 (96%) | 7 (4%) | | 1 | |
| Active | 1167 (97%) | 34 (3%) | .527 | .8 | .3–1.7 |
| TK (score) | -.01 (1.00) | -.42 (1.13) | .0058 | .4 | .2–0.8 |
| LANDMARK (score) | .03 (0.98) | -.11 (0.80) | .054 | .8 | .6–1.1 |
| ROUTE (score) | .06 (0.96) | -.51 (0.97) | < .001 | .5 | .3–.7 |
| SURVEY (score) | .02 (0.99) | -.99 (0.70) | < .001 | .2 | .1–.3 |

*Chi square or Anova test

Note: Sub-totals are not 1,698 because some participants did not fill in some questions. RLC = Right-Left confusion; MoT = Means of Travelling; TK = Town Knowledge.

considered the percentage of the DTD in a sample of 1,698 participants, using as a basic criterion 2 standard deviations below the means of the SOD. Then, we also investigated the critical factors predictive of both the SOD and DTD. As in Iaria and Barton's study [74], we confirmed that the rate of occurrence of the disorder is not a rare condition; rather, it affects 3% of young people undermining their autonomy and ability to work away from family boundaries.

As concerns the predictive factors of SOD, the analyses showed that gender more than educational level is strictly related to SOD. In particular, although females use more landmark-based navigational strategies and complain more difficulties in SOD, males show a higher risk of DTD than females. Indeed, given a glance at the DTD literature, the most of the single cases described are males; this means that although males have better visuospatial and navigational skills than females, they are also the most fragile [69, 70, 72, 75, 77]. Moreover, SOD was related to TK and Survey competencies. Iaria and co-workers [87] clarified that normal navigators can switch from one strategy to another by increasing their familiarity with the environment. This means that it is possible to implement higher navigational skills by acting on environmental knowledge. Also in Nori and Piccardi [37] emerged that when the individual was familiar with the environment, even if s/he generally preferred low navigational strategies, s/he was able to perform more complex navigational tasks in that specific environment, therefore, citizens of Bologna who were familiar with a neighbourhood of the city were able to recognize rotated monuments of the city even though their ability to mentally rotate an object was low [37, 84].

Consistent with these results, we found that TK and the Survey strategy are negatively related to DTD, suggesting that they can be protective factors in counteracting the onset of

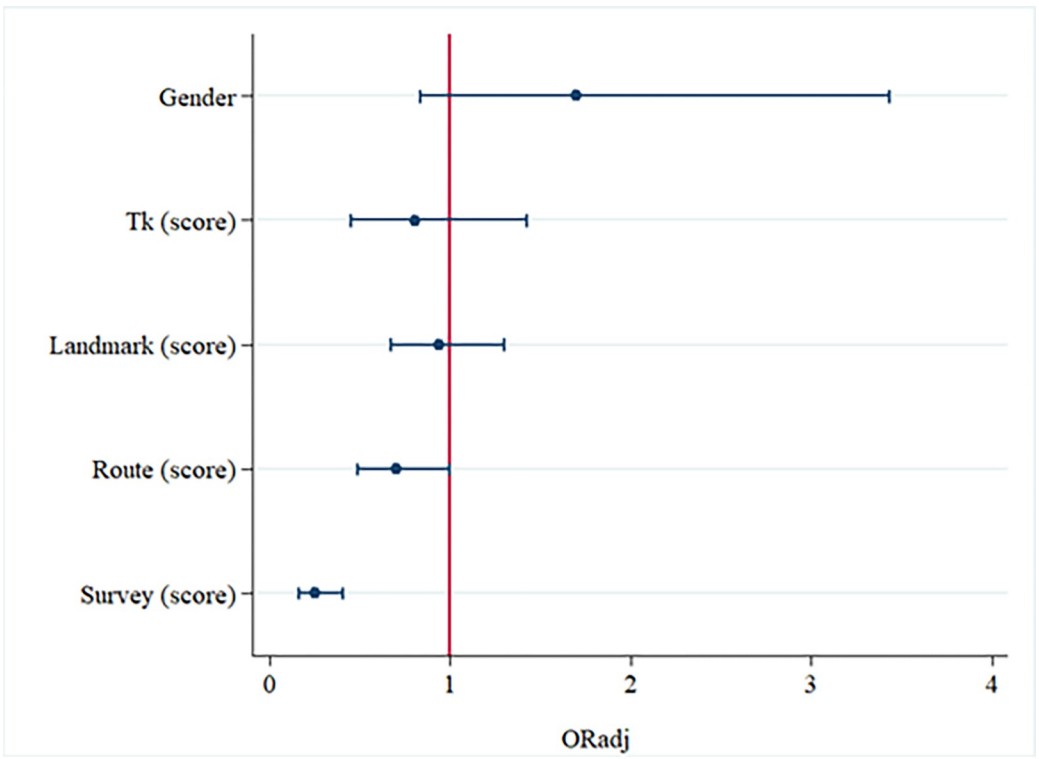

**Fig 2. Multivariable association between five risk factors and DTD.** Note: error bars are 95% confidence intervals.

DTD. In this vein, Bartonek and co-workers [88–90] showed that children with cerebral palsy and motor disability manifested differences in topographic working memory as a function of the degree of autonomy to explore the environment, regardless of motor impairment. Our results also showed that the use of advanced navigational strategies (survey strategies) is not associated with the presence of DTD. In line with this finding, we also showed that individuals who achieve high spatial skills do not complain of RLC, which is generally involved in navigational disorders [60]. Consistent with this result, Giancola et al. [36] showed that survey strategies also characterise samples of experienced navigators, such as military pilots. According to Verde et al. [32, 33, 91, 92] this professional category is already selected on the basis of spatial and navigational skills. In addition, the military training would reinforce the survey strategy even more, including all those cognitive processes related to navigation, such as the ability to mentally rotate two and three-dimensional objects [93] or the ability to make directional and metric judgments.

Undoubtedly, the presence of a navigational deficit can also make the subject anxious and more reluctant to explore the environment, so it is difficult to determine how much one comes before the other. This result is particularly interesting in light of Lopez et al.'s [94] study, which showed that the role of the direct experience with exploring hometown on spatial mental representations appeared to be more important in the elderly than in young people. Our sample includes young people, therefore we can imagine that TK and the Survey strategy protects not only seniors from the detrimental effects of ageing on spatial mental representations but also young people in acquiring spatial competence by reducing the risk of navigational disorders.

Given the key role of spatial strategies, our results imply navigational training starting with pre-schoolers, in order to prevent the DTD, such as the one already used in Boccia et al. [95],

which allows implementing spatial orientation and autonomy skills from the earliest years of life, starting in kindergarten. The introduction of navigational training in education settings may be useful not only for healthy children but also for children with different types of disabilities (e.g. sensory-motor impairments or acquired brain-damaged or ADHD: [96]), who show several navigational disorders. Furthermore, given that simply enhancing cognitive performance is insufficient to reduce a sense of inadequacy about one's ability [97], the introduction of training activities specifically designed to improve metacognition would improve self-efficacy in individuals with respect to their SOD and related activities, and by consequence would reduce the risk of DTD. In this vein, De Lisi and Wolford [98] showed children improvements in mental rotation through the daily practice of the popular video game Tetris. This policy of intervention could have important spin-offs increasing also social life.

The current research is not without limitations. First, the study was conducted using an online self-reported survey. In the future it will be important to investigate DTD in presence using a battery of navigational tests. For example, no significant result was found in terms of MoT (active and/or passive movements in the environment). Probably this aspect should be investigated differently by quantifying more precisely the movements and their duration and stratifying the sample by strategies developed. Future work should investigate this component along with environmental characteristics (size of the place where one lives; the presence of distant landmarks; need to travel far to receive medical care or to use school services). Second, individuals with DTD were not tested for other cognitive deficits using a battery of neuropsychological tests, but were only asked to report if they suffered from cognitive disorders. Third, the diagnosis of DTD was not supported by structural imaging data in order to exclude the presence of any brain lesions.

In conclusion, the present study allowed us to identify in a large sample of young Italians the presence of DTD and its occurrence. It has also allowed us to observe protective factors that are associated with good navigational skills and that in the future can be used within protocols for the prevention of the development of spatial orientation disorders, as well as to promote these skills by reducing the gender gap that still emerged in this sample.

## Author Contributions

**Conceptualization:** Laura Piccardi, Massimiliano Palmiero, Maddalena Boccia, Liana Palermo, Cecilia Guariglia, Raffaella Nori.

**Data curation:** Massimiliano Palmiero, Vincenza Cofini, Raffaella Nori.

**Formal analysis:** Vincenza Cofini.

**Funding acquisition:** Paola Verde.

**Investigation:** Massimiliano Palmiero, Paola Verde, Maddalena Boccia, Liana Palermo, Cecilia Guariglia.

**Methodology:** Laura Piccardi.

**Software:** Vincenza Cofini, Raffaella Nori.

**Supervision:** Laura Piccardi.

**Writing – original draft:** Laura Piccardi.

**Writing – review & editing:** Laura Piccardi, Massimiliano Palmiero, Vincenza Cofini, Paola Verde, Maddalena Boccia, Liana Palermo, Cecilia Guariglia, Raffaella Nori.

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
