## [Decision Letter · Decision Letter 0]

8 Mar 2022

PONE-D-22-02928Where am I? An Italian study on the prevalence in young adults of Developmental Topographical Disorientation.PLOS ONE

Dear Dr. Piccardi,

Thank you for submitting your manuscript to PLOS ONE. After careful consideration, we feel that it has merit but does not fully meet PLOS ONE’s publication criteria as it currently stands. Therefore, we invite you to submit a revised version of the manuscript that addresses the points raised during the review process. Please pay particular attention to my comments below.

We look forward to receiving your revised manuscript.

Kind regards,

David Giofrè, Ph.D.

Academic Editor

PLOS ONE

Journal Requirements:

- https://www.frontiersin.org/articles/10.3389/fphar.2017.00496/full

The text that needs to be addressed involves the Introduction.

In your revision ensure you cite all your sources (including your own works), and quote or rephrase any duplicated text outside the methods section. Further consideration is dependent on these concerns being addressed.

Additional Editor Comments:

Dear dr. Piccardi,

I have now received comments from two reviewers, who are suggesting some more information and clarifications. Reviewer 1 suggested some minor changes, whereas comments from the reviewer 2 are substantial. There are several parts that were not entirely clear and that require some further clarifications, including missing information about your sample and materials.

I also had the opportunity to revise your manuscript, and I also have some suggestions as well.

Generally, please omit the 0 from correlations and from p-values

Table 1. N of males 635 + 1,062 = 1697, one participant is missing and this should be specified.

Line 214 “Spearman correlation analysis”, which is fine, but the authors might also want to explain why they decided to use rank order correlations.

Line 215, to the one hand a non-parametric approach was used, but few lines below generalized linear mixed models were used, which assume linearity. I think that some clarification is needed here.

Line 229 “multivariable logistic model”, this is ok, but I would probably suggest to include more information about this model.

Line 225 and subsequent (3.87±4) I assume that those are SDs, but this is not particularly clear and should be clarified.

Line 230 and subsequent, the statistical approach relies entirely on NHST, while correlations should also be interpreted in terms of magnitude.

Line 235 “and it was not correlated” this is statistically inaccurate, not statistically significant does not necessarily mean no correlation. Please also note that (rho=-0.045; p=0.066) is statistically significant if one tail. Here again the author might want to discuss the correlation in terms of magnitude, .045 is particularly small.

Line 238. Table 1 should be Table 2 (please amend all subsequent tables). Please also report the degrees of freedom for each correlation if the N is different across different measures.

Line 240 “Multivariable analysis” more information are required here, which multivariable analysis? Which kind of regression has been used here? Why interaction terms have been included, is there a rationale for this inclusion? Also, please report CIs and standardized betas for the regression model.

Line 247 “the prevalence of DTD was 3% (54/168:…” Is this accurate? 54/168 = 32.14%

Line 248 "The univariate analysis showed that DTD was associated with gender: the males had a higher risk than females (OR: 2.39; 95%CI:1.2-4.7)," which univariate analysis? Are those logistic regressions? If so, why did you decide to perform univariate analyses rather than including everything in the same model? I think that this might be ok, but some clarification is needed here.

Line 256 “Multivariable analysis” here again please provide some more information.

Line 256 “Survey (ORadj=” Please also clarify why you decided to used adjusted values.

Line 254 “Totals are not 1,698 because of missing data” this is quite unclear and should be clarified.

Figure 2. I guess that error bars are confidence intervals but this should be specified.

Reviewers' comments:

Reviewer's Responses to Questions

**Comments to the Author**

1. Is the manuscript technically sound, and do the data support the conclusions?

Reviewer #1: Yes

Reviewer #2: Partly

2. Has the statistical analysis been performed appropriately and rigorously? 

Reviewer #1: Yes

Reviewer #2: Yes

3. Have the authors made all data underlying the findings in their manuscript fully available?

Reviewer #1: Yes

Reviewer #2: No

4. Is the manuscript presented in an intelligible fashion and written in standard English?

Reviewer #1: Yes

Reviewer #2: Yes

5. Review Comments to the Author

Reviewer #1: The ms. entitled "Where am I? An Italian study on the prevalence in young adults of Developmental

Topographical Disorientation" is interesting and well written.

Anyway I suggest some integrations as follows.

Review the introduction:

I suggest to add some information regarding how the individuals encode categorical and coordinate spatial information (please see 10.1111/sjop.12633; 10.1016/j.jenvp.2020.101392), and their role in the development of the disorder.

Review the method:

Line 171 pg 8 "All participants gave their informed consent before their inclusion in the study" seems to me a repetition

Please add a table with the items of Familiarity and Spatial Cognitive Style Scale as supplementary materials.

I have not concernes about statistical analysis and results.

Review the Discussion:

Please clarify this sentence:"For such a reason, preferred navigational strategies may not be attributed exclusively to

environmental characteristics, even if they may affect the development of navigational strategies".

Please add the limitations of the study.

Please check some typos throughout the manuscript.

Reviewer #2: 1,698 fill out a questionnaire investigating demographic characteristics, sense of direction, spatial cognitive styles, city of residence knowledge, left-right confusion, way of travelling.

They found that 3% of people met the criteria for Developmental Topographical Disorientation

I think the paper addresses an interesting topic. However, the introduction lack of a coherent line of reasoning, the rationale for the study aims, and material is not completely clear.

Specific points:

INTRODUCTION

Row 49. “Sense of Direction (SOD) is the ability”. I would speak of “Navigation is the ability”. Sense of direction is an indirect measure, it is our own perception about navigation ability.

Row 62. Why are internal factors of greater interest in the present study?

Row 64. I suggest dividing “professional experience” and “familiarity with the environment”. They are both individual factors associated with navigation ability, but I would consider them two distinct factors.

Rows 88-92. I found this paragraph confusing. Did you mean to introduce the “familiarity” with the environment factor? If so, I suggest rephrasing and giving more explanation of the point. If not, please make this paragraph clearer and linked with the line of reasoning.

AIMS AND HYPOTHESES

Row 149. The first aim: “(i) to investigate the Sense of Direction (SOD) and its correlates” is not clear to me. I think this was not introduced well from the title, to the abstract, and to the literature.

Did you mean you wanted to investigate people demographic characteristics, knowledge about their environment, cognitive style with their self-reported sense of direction? And which is the rationale of this choice? Why not considering spatial cognitive abilities for instance.

Furthermore, if this was the aim, the presentation of previous findings on the association between familiar environment knowledge and other factors with SOD merit to be better introduced.

Hypotheses are completely missing. What were you expecting?

Row 155. “Study Design It was a national online survey on the DTD.” What do you mean with national?

PARTICIPANTS

You spoke about prevalence, but is your sample representative of the young population?

I think more information about sample characteristics is necessary.

For instance

- Years of education. Are they all university students? I suggest adding more information on this topic.

- Recruitment. How was the sample recruited? You stated “awareness campaign”. But are many of them university (psychology) students?

- The provenance of participants? You briefly stated this in discussion, I think information should be added in Participants section

MATERIAL

You used a questionnaire asking about demographic variables, sense of direction, their own city of residence knowledge, spatial cognitive styles, left-right confusion. Then you mentioned also the way of travelling.

I am missing some points, and I think this deserves to be more clearly presented.

- What are the Cronbach alphas of the current sample?

- Does the sense of direction refer to both new and familiar environments?

- City knowledge also included how many years they lived in?

- Which is the rationale to measure left-right confusion? And ways of travelling? They are not presented in the introduction.

- Information about scoring? (Figure 1 depict a 1-5 point scale)

Row 235. “disorientation”. What do you mean?

Row 247. I am missing a point, what are the criteria?

Rows 300-305 Why are you referring to older adults? Please make this point clearer

6. PLOS authors have the option to publish the peer review history of their article (what does this mean?). If published, this will include your full peer review and any attached files.

Reviewer #1: No

Reviewer #2: No

---

## [Author Response · Author response to Decision Letter 0]

19 May 2022

In the manuscript, all changes are highlighted in yellow.

PONE-D-22-02928

Where am I? An Italian study on the prevalence in young adults of Developmental Topographical Disorientation.

PLOS ONE

Dear Dr. Piccardi,

Thank you for submitting your manuscript to PLOS ONE. After careful consideration, we feel that it has merit but does not fully meet PLOS ONE’s publication criteria as it currently stands. Therefore, we invite you to submit a revised version of the manuscript that addresses the points raised during the review process. Please pay particular attention to my comments below.

We look forward to receiving your revised manuscript.

Kind regards,

David Giofrè, Ph.D.

Academic Editor

PLOS ONE

Journal Requirements:

- https://www.frontiersin.org/articles/10.3389/fphar.2017.00496/full

The text that needs to be addressed involves the Introduction.

In your revision ensure you cite all your sources (including your own works), and quote or rephrase any duplicated text outside the methods section. Further consideration is dependent on these concerns being addressed.

Revisiting the introduction following the Reviewers’ comments we solved the issue about of the overlapping text. We also checked the duplicated text outside the method section. 

Additional Editor Comments:

Dear dr. Piccardi,

I have now received comments from two reviewers, who are suggesting some more information and clarifications. Reviewer 1 suggested some minor changes, whereas comments from the reviewer 2 are substantial. There are several parts that were not entirely clear and that require some further clarifications, including missing information about your sample and materials.

I also had the opportunity to revise your manuscript, and I also have some suggestions as well.

Generally, please omit the 0 from correlations and from p-values

Done.

Table 1. N of males 635 + 1,062 = 1697, one participant is missing and this should be specified.

We apologize for this, there was a distraction error. We reported the correct number of females.

Line 214 “Spearman correlation analysis”, which is fine, but the authors might also want to explain why they decided to use rank order correlations.

Thank you for your suggestion. We tried to better report the choice of the correlation’s coefficients used. Our choice is related to the type of the investigated variables. Some of them in fact are binary or ordinal variables and other continuous. We used the Spearman's rank-order correlation, that is the nonparametric version of the Pearson product-moment correlation, to analyze the relationships between two variables measured on at least an ordinal scale. We specify the choice on the manuscript, also reporting that we used the point-biserial correlation to investigate the correlation between binary variables (the results were the same using Spearman).

Line 215, to the one hand a non-parametric approach was used, but few lines below generalized linear mixed models were used, which assume linearity. I think that some clarification is needed here.

Please, as reported above we explored the relationship between SOD and the other variables using non-parametric approaches (correlations) with respect to the type of the variables (continuous, ordinal and categorical). However, please consider that we did not use the generalized linear mixed models. In order to assess which independent variable to enter into the model used (Generalized Linear Model), we investigated the relation between the SOD variable and the others variables using the p-values criterion (p<0.05). “Generalized Linear Model” is a generalization of the general linear model. The general linear model assumes linearity, whereas the relationship in the generalized linear model between dependent variable and independent variables can be non-linear [Generalized Linear Models and Extensions, Fourth Edition. James W. Hardin and Joseph M. Hilbe.STATA PRESS, 2018]. 

We used generalized linear model specifying an identity link function and a normal family distribution than it is equivalent to a (general) linear model, so we calculated the coefficients , adding all interactions with categorical factors (gender, travel and disorientation) because the interactions helped explaining more variability of Y (R2=.334 of the model with the main effects only vs R2 =.337 of the model with interactions).

We included this information in the manuscript.

Line 229 “multivariable logistic model”, this is ok, but I would probably suggest to include more information about this model.

Thank you for suggestion we reported it on the manuscript as you asked.

Line 225 and subsequent (3.87±4) I assume that those are SDs, but this is not particularly clear and should be clarified.

Thank you for your suggestion; we now reported standard deviations into parenthesis and we deleted the symbol ±. 

Line 230 and subsequent, the statistical approach relies entirely on NHST, while correlations should also be interpreted in terms of magnitude.

WE investigate the relation between the SOD variable and the others using the p-values (p<.05) to assess the statistical significance of the result to enter the independent variables into the glm model.

However, as suggested by the Reviewer we added information about the magnitude of correlations in the manuscript. 

Line 235 “and it was not correlated” this is statistically inaccurate, not statistically significant does not necessarily mean no correlation. Please also note that (rho=-0.045; p=0.066) is statistically significant if one tail. Here again the author might want to discuss the correlation in terms of magnitude, .045 is particularly small.

Thank you for your comment. We agree with you that not statistically significant does not necessarily mean no correlation, and the correlation should be discussed in terms of magnitude, but we used the p-values approach to choose the independent variables in our model. All tests were two-sided, we reported it on the manuscript

However, we included in the manuscript both significance and the magnitude issues, in the correlation section. 

Line 238. Table 1 should be Table 2 (please amend all subsequent tables). Please also report the degrees of freedom for each correlation if the N is different across different measures.

Thank you, We reported consecutive numbers for all tables, and we reported n and degrees of freedom for each correlation in table 2

Line 240 “Multivariable analysis” more information are required here, which multivariable analysis? Which kind of regression has been used here? Why interaction terms have been included, is there a rationale for this inclusion? Also, please report CIs and standardized betas for the regression model.

We have integrated the information about glm as you requested in method’s section reporting standardized beta and CIs on table 3 and specifying the type of regression model in the manuscript. 

Line 247 “the prevalence of DTD was 3% (54/168:…” Is this accurate? 54/168 = 32.14%

We apologize for this, the percentage is 3%: 54/1698, there was a typing error. We have corrected it.

Line 248 "The univariate analysis showed that DTD was associated with gender: the males had a higher risk than females (OR: 2.39; 95%CI:1.2-4.7)," which univariate analysis? Are those logistic regressions? If so, why did you decide to perform univariate analyses rather than including everything in the same model? I think that this might be ok, but some clarification is needed here.

Thank you for your suggestions, we revised the method’s section reporting more information about it. First, we used logistic univariate regression, to reduce the number of independent variables to include in our model. We included in the multiple logistic model only the factors (predictors) that were statistically significant (p<0.05 and Odds ratio different from 1, that is called the “crude OR” or unadjusted OR). An adjusted odds ratio is an odds ratio that has been adjusted to account for other predictor variables in a model. 

Line 256 “Multivariable analysis” here again please provide some more information.

Thank you for your suggestion. We added it. 

Line 256 “Survey (ORadj=” Please also clarify why you decided to used adjusted values.

Thank you. We have attempted to clarify it in the text. 

Line 254 “Totals are not 1,698 because of missing data” this is quite unclear and should be clarified.

We clarified in the note of the table 4 that some subtotals are different from total count because some participants did not fill in all questions, which is why the subtotal for some variables is different. For correctness we reported this information.

Figure 2. I guess that error bars are confidence intervals but this should be specified.

Done, thank you.

Reviewer 2:

Review the introduction:

I suggest to add some information regarding how the individuals encode categorical and coordinate spatial information (please see 10.1111/sjop.12633; 10.1016/j.jenvp.2020.101392), and their role in the development of the disorder.

We added the reviewer’s suggestion, including more information about individuals’ categorical and coordinate spatial encoding, as well as an interpretation of the role of these types of encoding in the development of the DTD. Indeed, in general, individuals with DTD show a higher impairment of metric (coordinates) than nonmetric (categorical) spatial encoding, and, basing on Siegel and White's model, they hardly reach a level of route knowledge of the environment, often stopping at a landmark knowledge, highlighting a lack of allocentric representation capacity.

Review the method:

Line 171 pg 8 "All participants gave their informed consent before their inclusion in the study" seems to me a repetition

Done

Please add a table with the items of Familiarity and Spatial Cognitive Style Scale as supplementary materials.

The full items of Familiarity and Spatial Cognitive Style Scale are already published in the Appendix of the paper “Piccardi, L., Risetti, M., & Nori, R. (2011). Familiarity and environmentalrepresentations of a city: a self-report study. Psychological reports, 109(1), 309-326.” Therefore, wecouldnotreproduceitagain, butwe indicate this in the method.

I have not concernes about statistical analysis and results.

Review the Discussion:

Please clarify this sentence:"For such a reason, preferred navigational strategies may not be attributed exclusively to environmental characteristics, even if they may affect the development of navigational strategies".

We dropped the sentence because we realized that it was not clear and confounded the reading of the manuscript.

Please add the limitations of the study.

We added the limitations of the study as follows, before the last paragraph: ‘The current research is not without limitations. First, the study was conducted using an online self-reported survey. In the future it will be important to investigate DTD in presence using a battery of navigational tests. Second, individuals with DTD were not tested for other cognitive deficits using a battery of neuropsychological tests, but were only asked to report if they suffered from cognitive disorders. Third, the diagnosis of DTD was not supported by structural imaging data in order to exclude the presence of any brain lesions’. 

Please check some typos throughout the manuscript.

We checked and corrected all typos throughout the manuscript. 

Reviewer #2:

I think the paperaddresses an interestingtopic. However, the introductionlack of a coherent line of reasoning, the rationale for the studyaims, and materialisnotcompletelyclear.

Following the Reviewers’ directionswefullyreviseted the introduction and the rationale and weclarified the material.

Specific points:

INTRODUCTION

Row 49. “Sense of Direction (SOD) is the ability”. I would speak of “Navigation is the ability”. Sense of direction is an indirect measure, it is our own perception about navigation ability.

We corrected this point and specified that SOD is an indirect measure of the navigational ability. 

Row 62. Why are internal factors of greater interest in the present study?

We specified at the beginning of the introduction that internal factors are important because they affect navigational competence; by consequence it is possible to better intervene on them in order to plan prevention programs related to navigational disorders. We then clarified which internal factors were used for our study, namely demographic factors (e.g., age, gender and education), the degree of familiarity with the environment (e.g., town knowledge), and navigational strategies, which include not only cognitive styles (e.g., landmark, route and survey), but also the preferential mean of movement (active, passive). In addition, we also considered the right-left confusion (RLC) as a pathological factor that can be associated or not with navigational disorders.

Row 64. I suggest dividing “professional experience” and “familiarity with the environment”. They are both individual factors associated with navigation ability, but I would consider them two distinct factors.

As suggested by the reviewer, when listing the internal factors, we divided ‘professional experience’ from ‘familiarity with the environment’, supporting the two factors with specific bibliography. 

Rows 88-92. I found this paragraph confusing. Did you mean to introduce the “familiarity” with the environment factor? If so, I suggest rephrasing and giving more explanation of the point. If not, please make this paragraph clearer and linked with the line of reasoning.

Well, no, we refer to familiarity still as an internal factor. Indeed, following Craig et al. (2012, p. 2), ‘Intuitively, familiarity is simply the result of repeated exposure to a particular stimulus or environment. Indeed, this intuitive interpretation of familiarity is the basis of the mere-exposure effect put forward by Zajonc [10], which theorises that preference for objects can be induced by repeated exposure. This kind of familiarity might be called (after Zajonc), “objective familiarity,” or “actual familiarity”—a simple correlate of the number of times a person has seen a particular object or scene’. Then, we corrected the sentence as follows in order to avoid confusion: ‘Montello [5] reported that familiarity is more associated with a survey format (similar to cognitive maps)’. In addition, given the general restructuration of the introduction, as suggested in a previous comment, we better connected the paragraph to the rest of the text. 

AIMS AND HYPOTHESES

Row 149. The first aim: “(i) to investigate the Sense of Direction (SOD) and its correlates” is not clear to me. I think this was not introduced well from the title, to the abstract, and to the literature.

Did you mean you wanted to investigate people demographic characteristics, knowledge about their environment, cognitive style with their self-reported sense of direction? And which is the rationale of this choice? Why not considering spatial cognitive abilities for instance.

Furthermore, if this was the aim, the presentation of previous findings on the association between familiar environment knowledge and other factors with SOD merit to be better introduced.

Thank to the Reviewer’s observation, we clarified the objective,which is the investigation of DTD, and accordingly we introduced the hypotheses. Concerning spatial abilities, we did not test them through paper and pencil tests because we were interested to detect DTD population, which is assessed using the Sense of Direction measure, and not by spatial tasks measuring spatial abilities.As described in the paper the DTD is measured considering 2 standard deviations below the mean of the sense of direction. Nevertheless,we believe that assessing spatial tasks in DTD deserves further studies. 

Hypotheses are completely missing. What were you expecting?

We thank Reviewer for his/her suggestions, we now add at the end of the introduction the objectives also the hypotheses.Specifically, our objective and hypotheses are related to the DTD. 

Row 155. “Study Design It was a national online survey on the DTD.” What do you mean with national?

Thank you for this comment, we meant a survey conducted by the Italian Population. We corrected this point also in the manuscript.

PARTICIPANTS

You spoke about prevalence, but is your sample representative of the young population?

Thanks to the Reviewer’s suggestion we decided to substitute the term ‘prevalence’ with the term ‘percentage’, which is more correct considering the size of our sample. 

I think more information about sample characteristics is necessary.

For instance

- Years of education. Are they all university students? I suggest adding more information on this topic. 

We specified in the manuscript the % of their educational levels as follows: Participants had a full-time education, ranging from 8 to 18 years (mean = 14.80 years, SD = 2.83 years). This means that they were not all university students. Specifically, 4.79% of the participants achieved a secondary school diploma, 49.76% achieved a high school diploma, and 45.45% achieved a degree or post-degree education level. 

- Recruitment. How was the sample recruited? You stated “awareness campaign”. But are many of them university (psychology) students?

Participants were recruited through notices on social networks and on bulletin boards of researchers. We specified that ‘the advices about the survey were basically spread out by word of mouth and flyers, that were distributed in community meeting points, such as bookshops, cafeterias, public library, and sport clubs’.

The provenance of participants? You briefly stated this in discussion, I think information should be added in Participants section

We recruited participants from all Italian regions, including the Islands. We specified it in the participants’ section. 

MATERIAL

You used a questionnaire asking about demographic variables, sense of direction, their own city of residence knowledge, spatial cognitive styles, left-right confusion. Then you mentioned also the way of travelling.

In the same questionnaire along with demographic characteristics, the participants were also asked how they moved around the environment, whether they used active or passive means of transport. Specifically, we investigated the use of means of transport, distinguishing active means of transport in which the participant actively drives and moves around the environment and passive means of transport in which the participant is led around the environment by others. We have replaced the term travel with active and/or passive means of transport to avoid misunderstandings.

I am missing some points, and I think this deserves to be more clearly presented.

- What are the Cronbach alphas of the current sample?

We reported the Cronbach’s alpha for the total scale for the sample we used, and it was = .71. 

- Does the sense of direction refer to both new and familiar environments?

Yes, it refers to both new and familiar environments. We specified it in the manuscript the first time we introduced the concept of SOD.

- City knowledge also included how many years they lived in?

When we refer to the city (town) knowledge, all questions concern a town that the subject knows well. In fact, at the beginning of the questionnaire they were expressly asked to imagine a very familiar town and to indicate its name, while for the SOD the questions concern more or less familiar places. 

- Which is the rationale to measure left-right confusion? And ways of travelling? They are not presented in the introduction.

Regarding the right/left confusion, the rational is that it is one of the visuo-spatial disorders and in our case is a kind of anamnestic question because we believe that this disorder affects the ability to verbally label left and right correctly but does not have an effect on the mental representation of space.

The ways of traveling, instead, are associated with the ability to orient oneself in space, which improves with practice. Indeed, active travel often indicates a better sense of direction and leads to a better representation of one's surroundings. In general, people with less navigation ability tend to move less and move with others in a passive manner. 

We included this rationale in the introduction. 

- Information about scoring? (Figure 1 depict a 1-5 point scale)

We reported in the ‘data collection’ section that ‘for each item, participants should circle a number from 1 to 5 to indicate their response: higher numbers correspond to higher ability. Then, the final scores were the means of the item scores’. Thus, the scoring involved the average of the item scores for each subscale (route, survey, landmark, SOD and TK). 

Row 235. “disorientation”. What do you mean?

We mean ‘right-left confusion’. We Specified it throughout the manuscript.

Row 247. I am missing a point, what are the criteria?

We thank Reviewer for his/her comment. Indeed, we had not been clear in making explicit the diagnostic criteria for the disorder. Now, in addition to pointing out what score the subjects had to obtain in order to be considered affected by DTD, we have explicitly listed the international criteria (Iaria and Barton 2010) for the disorder and the two additional criteria that we have considered in this paper.

Rows 300-305 Why are you referring to older adults? Please make this point clearer

By making revisions to the manuscript, this point is no longer present.

---

## [Decision Letter · Decision Letter 1]

21 Jun 2022

PONE-D-22-02928R1“Where am I?” A snapshot of the Developmental Topographical Disorientation among young Italian adults.PLOS ONE

Dear Dr. Piccardi,

Thank you for submitting your manuscript to PLOS ONE. After careful consideration, we feel that it has merit but does not fully meet PLOS ONE’s publication criteria as it currently stands. Therefore, we invite you to submit a revised version of the manuscript that addresses the points raised during the review process.

We look forward to receiving your revised manuscript.

Kind regards,

David Giofrè, Ph.D.

Academic Editor

PLOS ONE

Journal Requirements:

Additional Editor Comments:

Dear Dr. Piccardi,

you can see that the reviewers have now commented on your paper. One reviewer suggests accepting the paper as it stands now, while the other raises some concerns. By my own reading of the paper, I noticed that the manuscript has greatly improved. Therefore, I am encouraging you to revise the paper according to the instructions provided by the second reviewer.

Best wishes,

David Giofrè

Reviewers' comments:

Reviewer's Responses to Questions

**Comments to the Author**

1. If the authors have adequately addressed your comments raised in a previous round of review and you feel that this manuscript is now acceptable for publication, you may indicate that here to bypass the “Comments to the Author” section, enter your conflict of interest statement in the “Confidential to Editor” section, and submit your "Accept" recommendation.

Reviewer #1: All comments have been addressed

Reviewer #2: (No Response)

2. Is the manuscript technically sound, and do the data support the conclusions?

Reviewer #1: Yes

Reviewer #2: Partly

3. Has the statistical analysis been performed appropriately and rigorously? 

Reviewer #1: Yes

Reviewer #2: Yes

4. Have the authors made all data underlying the findings in their manuscript fully available?

Reviewer #1: Yes

Reviewer #2: Yes

5. Is the manuscript presented in an intelligible fashion and written in standard English?

Reviewer #1: Yes

Reviewer #2: Yes

6. Review Comments to the Author

Reviewer #1: The authors replied all the questions raised. The manuscript can be accepted in the present form.

Reviewer #2: The authors have addressed almost all my previous concern sufficiently well.

I have a few additional comments.

- I found the description of the questionnaires a bit confusing. You present them as part of a unique “familiarity and spatial cognitive style scale” questionnaire (pages 8-9). If I am understanding, this questionnaire was used as a total score in other studies, while here you considered the SOD part as your dependent variable, and town knowledge, cognitive styles, left/right confusion, and means of transport as your predictors.

Therefore, I suggest presenting the materials as follows

A) Anamnesis Questionnaire.

B) Demographic questionnaire (age, gender and educational level)

C) Means of transport questions (they are not numbered items of the scale, so I suppose they can be described as separately)

D) And then the “familiarity and spatial cognitive style scale” investigating in the order: sense of direction, left/right confusion, town knowledge, spatial cognitive styles.

- And for each part include all the information together (e.g., for the means of transport, you stated the items below in the text, but I would organize a dedicated paragraph in which finding all information about the measure, including scoring)

- For the Anamnesis Questionnaire could you add more explicitly what you asked the participants?

- Cronbach’s alphas of the subscales (SOD, town knowledge, spatial cognitive styles, ..) should be reported given the total score is not used here.

- Which is the rationale of considering the right/left dichotomous given data are collected in a 5-point Likert scale? Did the analyses differ when considering it not dichotomic?

- Familiarity. In the first revision, I pointed out that “Furthermore, if this was the aim, the presentation of previous findings on the association between familiar environment knowledge and other factors with SOD merit to be better introduced.”

But I do not think the authors have addressed this point. Familiarity is only described as part of the Siegel and White model to reach survey knowledge, but neither specific rationale (supported by literature) for measuring it and hypothesis are presented.

Specifically, why consider town knowledge as a predictor of SOD? This could be ok, if supported.

But I was also wondering, why not add the town knowledge score together with SOD, given that DTD involves impaired orientation both in familiar and unfamiliar environments? (as you stated: “In general, people with DTD have normal memory and neuropsychological profiles, but show a major cognitive deficit in spatial cognition and complain of severe problems in navigation on an everyday basis. Specifically, they are unable to use cognitive maps or place-based navigation strategies to find their way around familiar and novel environments”).

7. PLOS authors have the option to publish the peer review history of their article (what does this mean?). If published, this will include your full peer review and any attached files.

Reviewer #1: No

Reviewer #2: No

---

## [Author Response · Author response to Decision Letter 1]

25 Jun 2022

In the manuscript, all changes are highlighted in yellow.

PONE-D-22-02928R1

“Where am I?” A snapshot of the Developmental Topographical Disorientation among young Italian adults.

Journal Requirements:

Reply: the reference list has been checked.

Additional Editor Comments:

Dear Dr. Piccardi,

you can see that the reviewers have now commented on your paper. One reviewer suggests accepting the paper as it stands now, while the other raises some concerns. By my own reading of the paper, I noticed that the manuscript has greatly improved. Therefore, I am encouraging you to revise the paper according to the instructions provided by the second reviewer.

Best wishes,

David Giofrè

Reviewers' comments:

Reviewer's Responses to Questions

Comments to the Author

1. If the authors have adequately addressed your comments raised in a previous round of review and you feel that this manuscript is now acceptable for publication, you may indicate that here to bypass the “Comments to the Author” section, enter your conflict of interest statement in the “Confidential to Editor” section, and submit your "Accept" recommendation.

Reviewer #1: All comments have been addressed

Reviewer #2: (No Response)

2. Is the manuscript technically sound, and do the data support the conclusions?

Reviewer #1: Yes

Reviewer #2: Partly

3. Has the statistical analysis been performed appropriately and rigorously?

Reviewer #1: Yes

Reviewer #2: Yes

4. Have the authors made all data underlying the findings in their manuscript fully available?

Reviewer #1: Yes

Reviewer #2: Yes

5. Is the manuscript presented in an intelligible fashion and written in standard English?

Reviewer #1: Yes

Reviewer #2: Yes

6. Review Comments to the Author

Reviewer #1: The authors replied all the questions raised. The manuscript can be accepted in the present form.

Reviewer #2: The authors have addressed almost all my previous concern sufficiently well.

I have a few additional comments.

- I found the description of the questionnaires a bit confusing. You present them as part of a unique “familiarity and spatial cognitive style scale” questionnaire (pages 8-9). If I am understanding, this questionnaire was used as a total score in other studies, while here you considered the SOD part as your dependent variable, and town knowledge, cognitive styles, left/right confusion, and means of transport as your predictors.

Reply: We would like to thank the Reviewer who allowed us to clarify the description of the instrument in the manuscript. First, we substituted the word “scale” for “questionnaire” and “subscales” instead of “sections”. We also add some more information about MoT and reliability of the subscales as asked by the Reviewer in the following concerns. Concerning the scale is always administered in its entirety and provides a total score, but it also provides scores for SOD; town knowledge; right/left confusion and spatial cognitive styles which were particularly useful in the present work to better describe the navigational skills of the population under investigation. For this reason, we considered these aspects in the analyses. They have also been taken into account in other works (e.g., Piccardi et al 2011; Nori and Piccardi 2012), while in other studies only SOD was considered in order to exclude the presence of individuals with navigational disorders from the sample.

Therefore, I suggest presenting the materials as follows

A) Anamnesis Questionnaire.

B) Demographic questionnaire (age, gender and educational level)

C) Means of transport questions (they are not numbered items of the scale, so I suppose they can be described as separately)

D) And then the “familiarity and spatial cognitive style scale” investigating in the order: sense of direction, left/right confusion, town knowledge, spatial cognitive styles.

- And for each part include all the information together (e.g., for the means of transport, you stated the items below in the text, but I would organize a dedicated paragraph in which finding all information about the measure, including scoring)

Reply: We have now divided the instruments into A (Anamnesis Questionnaire) and B (familiarity and spatial cognitive style scale). With respect to demographic information and means of transport are part of the scale and cannot be described separately. However, we have added information to help the reader better understand how they are articulated, then in the text we also refer to the scale that is in the appendix of the work of Piccardi et al 2011, which is the reason why we cannot include it in the supplementary materials of this work. We hope that the added information can make the scale clearer.

- For the Anamnesis Questionnaire could you add more explicitly what you asked the participants?

Reply: We add some more information about questions we asked participants

- Cronbach’s alphas of the subscales (SOD, town knowledge, spatial cognitive styles, ..) should be reported given the total score is not used here.

Reply: We now provided the internal consistencies of SOD and TK to compare the values with the original studies (i.e. Piccardi, Risetti and Nori 2011). Both in Nori and Piccardi (2012) and Piccardi, Risetti and Nori (2011) the only reliability measures concerned the total score and TK and SOD, but not the other subscales, for this reason, lacking previous comparative values, we did not proceed to report the internal consistency of spatial cognitive styles.

- Which is the rationale of considering the right/left dichotomous given data are collected in a 5-point Likert scale? Did the analyses differ when considering it not dichotomic?

Reply: We checked the Spearman correlation between the right/left confusion and SOD using RLC as a continuous variable, and we found a similar result (Rho= -.12; p< .001) obtained using dichotomic classification. This let us to assume that also the subsequent analyses reported in the paper yield similar results. However, we have decided to transform RLC from continuous to dichotomic because representing the presence/absence of a disorder the dichotomic subdivision makes stronger the result.

- Familiarity. In the first revision, I pointed out that “Furthermore, if this was the aim, the presentation of previous findings on the association between familiar environment knowledge and other factors with SOD merit to be better introduced.”

But I do not think the authors have addressed this point. Familiarity is only described as part of the Siegel and White model to reach survey knowledge, but neither specific rationale (supported by literature) for measuring it and hypothesis are presented.

Specifically, why consider town knowledge as a predictor of SOD? This could be ok, if supported.

Reply: We thank Reviewer for raising this point that we missed. We now explained better the reason why we insert TK as predictor, supporting our hypothesis with literature and a theoretical model. 

But I was also wondering, why not add the town knowledge score together with SOD, given that DTD involves impaired orientation both in familiar and unfamiliar environments? (as you stated: “In general, people with DTD have normal memory and neuropsychological profiles, but show a major cognitive deficit in spatial cognition and complain of severe problems in navigation on an everyday basis. Specifically, they are unable to use cognitive maps or place-based navigation strategies to find their way around familiar and novel environments”).

Reply: We keep SOD and TK separate because they represent two independent albeit closely related constructs, i.e. it is true that increased familiarity with the environment produces positive effects on orientation, but as the Reviewer rightly points out in individuals with DTD the disorientation also occurs in highly familiar environments. It is now reasonable to see whether it can also represent a protective effect for them as we have now pointed out in the working hypothesis. Combining the two constructs, however, risks losing important information.

---

## [Decision Letter · Decision Letter 2]

29 Jun 2022

“Where am I?” A snapshot of the Developmental Topographical Disorientation among young Italian adults.

PONE-D-22-02928R2

Dear Dr. Piccardi,

We’re pleased to inform you that your manuscript has been judged scientifically suitable for publication and will be formally accepted for publication once it meets all outstanding technical requirements.

Kind regards,

David Giofrè, Ph.D.

Academic Editor

PLOS ONE

Additional Editor Comments (optional):

Reviewers' comments:

Reviewer's Responses to Questions

**Comments to the Author**

1. If the authors have adequately addressed your comments raised in a previous round of review and you feel that this manuscript is now acceptable for publication, you may indicate that here to bypass the “Comments to the Author” section, enter your conflict of interest statement in the “Confidential to Editor” section, and submit your "Accept" recommendation.

Reviewer #2: All comments have been addressed

2. Is the manuscript technically sound, and do the data support the conclusions?

Reviewer #2: Yes

3. Has the statistical analysis been performed appropriately and rigorously? 

Reviewer #2: Yes

4. Have the authors made all data underlying the findings in their manuscript fully available?

Reviewer #2: Yes

5. Is the manuscript presented in an intelligible fashion and written in standard English?

Reviewer #2: Yes

6. Review Comments to the Author

Reviewer #2: The paper has improved, the authors have addressed my previous issues and I have no more suggestions.

7. PLOS authors have the option to publish the peer review history of their article (what does this mean?). If published, this will include your full peer review and any attached files.

Reviewer #2: No

---

## [Editor Report · Acceptance letter]

7 Jul 2022

PONE-D-22-02928R2 

“Where am I?” A snapshot of the Developmental Topographical Disorientation among young Italian adults. 

Dear Dr. Piccardi:

I'm pleased to inform you that your manuscript has been deemed suitable for publication in PLOS ONE. Congratulations! Your manuscript is now with our production department. 

Kind regards, 

on behalf of

Dr. David Giofrè 

Academic Editor

PLOS ONE